# VSM: A Versatile Semi-supervised Model for Multi-modal Cell Instance Segmentation

**Xiaochen Cai**[1]
Caixc97@gmail.com

**Hengxing Cai**[2]
caihengxing@4paradigm.com

**Weiwei Tu**[2]
tuweiwei@4paradigm.com

**Kele Xu**[3]
kelele.xu@gmail.com

**Wu-Jun Li**[1,4][*]
liwujun@nju.edu.cn

1. National Key Laboratory for Novel Software Technology,
National Institute of Healthcare Data Science, Nanjing University, Nanjing, China
2. 4 Paradigm Inc., Beijing, China
3. School of Computer, National University of Defense Technology, Changsha, China
4. Center for Medical Big Data, Nanjing Drum Tower Hospital, Nanjing, China

## Abstract

Cell instance segmentation is a fundamental task in analyzing microscopy images, with applications in computer-aided biomedical research. In recent years, deep learning techniques have been widely used in this field. However, existing methods exhibit inadequate generalization ability towards multi-modal cellular images and require a considerable amount of manually labeled data for training. To overcome these limitations, we present VSM, a versatile semi-supervised model for multi-modal cell instance segmentation. Our method delivers high accuracy and efficiency, as verified through comprehensive experiments. Additionally, VSM achieved a top-five ranking in the Weakly Supervised Cell Segmentation category of the multi-modal High-Resolution Microscopy competition.

## 1 Introduction

Cell instance segmentation is a critical task in biological analysis and image processing applications, such as investigating intracellular processes, conducting single-cell quantitative analysis, and developing computer-aided cell physiology [1, 2, 3, 4, 5]. Unlike standard semantic segmentation [6, 7, 8] and object detection [9, 10, 11], instance segmentation requires simultaneous detection and segmentation of individual objects [12, 13]. Figure 1 provides an example of cell instance segmentation.

The impressive achievements of deep learning-based methods in the biomedical field [14, 15, 16, 17, 18] have led to an increasing interest in cell instance segmentation in microscopy images, with deep learning-based segmentation methods garnering significant research attention [19, 20]. Despite considerable efforts in this segmentation task, existing solutions still face significant challenges. The primary challenge lies in the diversity of image patterns and cell types [21, 22]. For example, blood cells in a bone marrow smear typically have rounded edges with a diameter of approximately 7 micrometers [23], while nerve cells have irregular edges and diameters [24]. Furthermore, obtaining

---

[*]corresponding author

36th Conference on Neural Information Processing Systems (NeurIPS 2022).

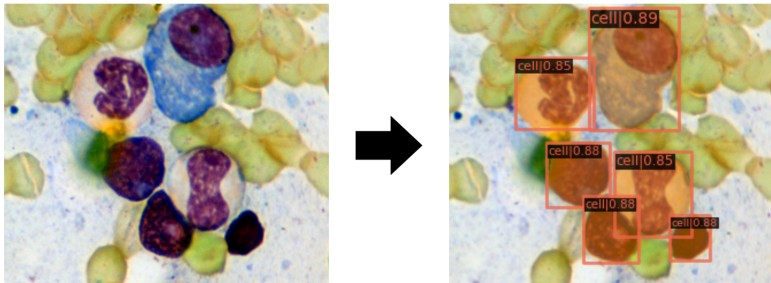

Figure 1: An example of cell instance segmentation, where pixels belonging to cells are labeled with an orange color.

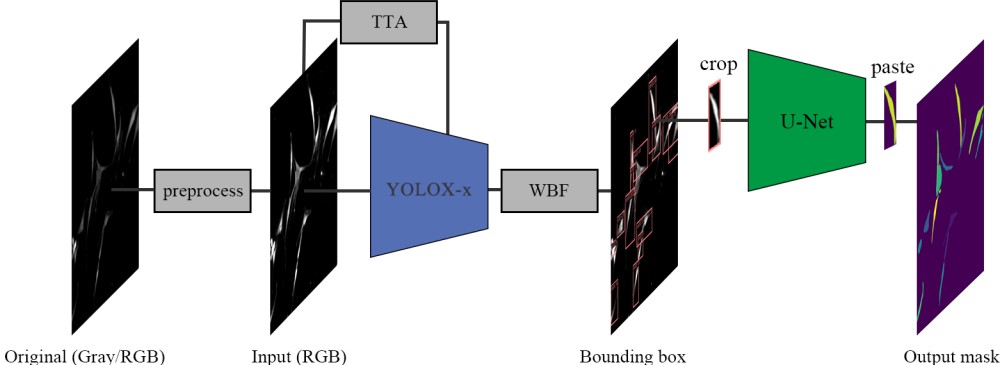

Figure 2: The overall pipeline of VSM.

a vast number of labeled cells across various modalities, including different imaging platforms and tissue types, is also a challenge. Existing methods are typically limited to a single type of image and lack generalization across modalities [2, 3, 18, 25]. Lastly, less attention is given to the utilization of readily available unlabeled data.

To address the aforementioned issues, we propose a versatile semi-supervised model (VSM) for cell instance segmentation. Specifically, VSM comprises a detection part to identify and locate individual cells in a microscopy image and a segmentation part to distinguish object and background for each cell bounding box. To maximize data utilization and overcome data limitations, VSM employs the following semi-supervised training [26] strategies: Firstly, VSM trains a model with limited labeled data, and the resulting model is referred to as the preliminary model, which is used to generate pseudo-labels [27] for unlabeled data through the application of a test-time-augmentation (TTA) strategy [28]. Secondly, VSM uses multiple data augmentation techniques, including flip and rotation, to amplify labeled data. VSM then mixes it with unlabeled data and pseudo-labels, and train another model utilizing the combined dataset, which outperforms the preliminary models. The effectiveness of VSM was authenticated in the Weakly Supervised Cell Segmentation in Multi-modality High-Resolution Microscopy Images competition, in which VSM achieved a 0.8535 F1 score on the tuning set and ranked among the top five.

The rest of the paper is organized as follows: Section 2 provides a detailed explanation of the methodology, and Section 3 presents the experimental settings. In Section 4, we present the experimental results. Finally, we conclude the paper in Section 5.

## 2  Methodology

The pipeline of VSM is illustrated in Figure 2. Note that VSM can accept various formats of cell microscopy images as inputs, including grayscale images, RGB images, and WSIs (whole slide images). The model directly outputs cell instance segmentation results without requiring manual intervention. We will present the details of each component of VSM in the following content.

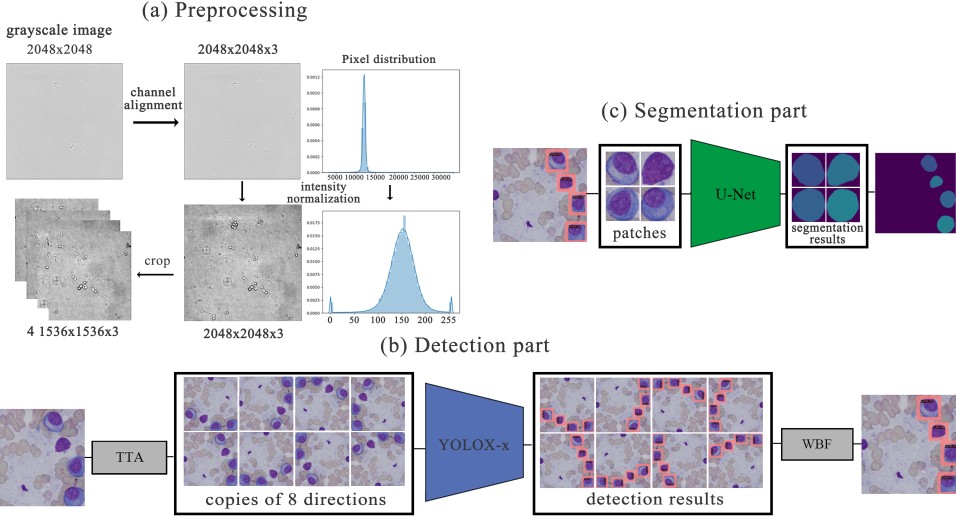

Figure 3: Each component of VSM.

## 2.1 Preprocessing

The preprocessing pipeline is illustrated in Figure 3(a). It comprises three steps: channel alignment, intensity normalization, and sliding cropping. We first perform channel alignment to ensure all images have the same number of channels. The gray images have only one channel, for which we replicate their channel for 3 times to make them have the same number of channels as that of RGB images. Then we normalize the pixel values of each channel to be between 0 and 255. The pixels with the top 1% value are set to 0, and the pixels with the last 1% value are set to 255. Finally, we crop images into patches with a window of $1536 \times 1536$ and a stride of 1024 to process large-resolution images and whole-slide images. We apply the same preprocessing pipeline in both training and inference.

## 2.2 Two-stage Instance Segmentation

Existing instance segmentation methods typically calculate object segmentation maps based on feature maps after feature extraction. However, these methods often ignore important low-level image information, such as edge, shape, and texture, which are essential in cell segmentation. Unlike natural images, cell images have simple semantic information and fixed structures, which necessitate equal emphasis on low-level and high-level semantic information in cell segmentation [29].

Therefore, we divide the task into two parts: the detection part and the segmentation part, as shown in Figure 3, where the detection part recognizes and locates every single cell with an object detection model, and the segmentation part segments the object and background from a small patch cropped from the original image. As aforementioned, the challenge of data limitation is a significant obstacle for multi-modal cell instance segmentation. To tackle the problem of the limitation of labeled data and take the most advantage of unlabeled data, we also develop a semi-supervised learning [26] pipeline to train the models with both labeled and unlabeled data.

### 2.2.1 Detection Part

Recently, object detection models have shown excellent generalization performance in complex scenes [30, 31, 32]. After preliminary experiment on the competition dataset, we find YOLOX [32] achieves good performance without hyper-parameter tuning. Furthermore, YOLOX [32] is an anchor-free model with a multistage detector, which is quite suitable for situations where the cell size varies greatly. Hence we take YOLOX as the cell detection network backbone. To further improve the proposals' quality, we apply the TTA strategy [28], where the input images are duplicated with 0, 90, 180, and 270 degrees of rotations and mirror rotations to obtain copies of the input image in eight directions. Here we use weighted box fusion [33] to merge the bounding box from different copies

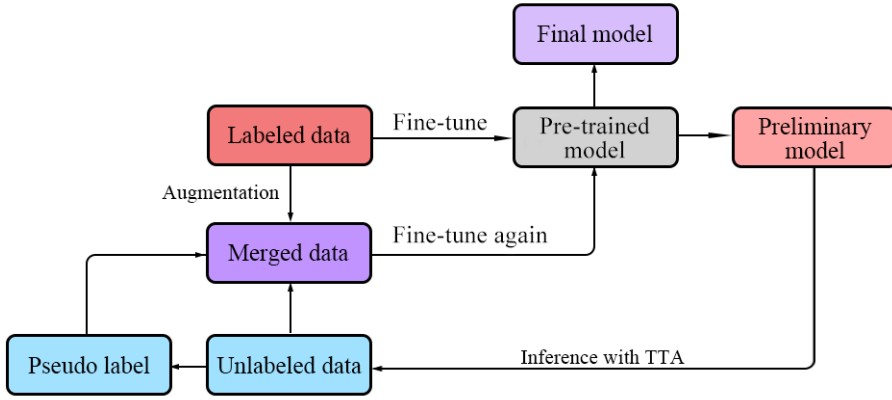

Figure 4: Semi-supervised training pipeline.

during development and replace it with Non-Maximum Suppression (NMS) during inference to speed up the inference phase. The workflow of the detection part is shown in Figure 3(b).

### 2.2.2 Segmentation Part

Cell microscopy images typically have a simple appearance and limited semantic information [29]. U-Net [7] can integrate low-level and high-level features during segmentation, making it a good choice for cell segmentation tasks with straightforward semantic information. Moreover, in our experiments, we found that the bottleneck of the inference speed of our model was the segmentation part. As a lightweight model, U-Net is helpful in improving inference efficiency. We tried to use segmentation models of better representation ability, but the experimental results showed that U-Net was a better choice in terms of both efficiency and accuracy (See Table 6). So We opted for U-Net as the segmentation network backbone. More specifically, we crop the image within the bounding box with confidence higher than 0.5 and resize it to $64 \times 64$ as model input, where bounding boxes with non-integer coordinates are rounded to the nearest. After segmentation operations, we restore the patch segmentation to its original size, assign a unique index and paste it into the segmentation mask of the whole image. The workflow of the segmentation part is shown in Figure 3(c).

### 2.2.3 Semi-supervised Learning

To address the challenge of the limitation of labeled data and take the most advantage of unlabeled data, we also develop a semi-supervised learning pipeline to train models with both labeled and unlabeled data, as shown in Figure 4. We first pre-train the model on the LIVECell dataset [34] and Sartorius competition data [35] with COCO pre-trained initialization. The pre-trained model is then fine-tuned on labeled data and is considered the preliminary model. It is used to generate pseudo-labels [27] for unlabeled data, during which a TTA strategy [28] is used. We apply multiple data augmentations, including flipping and rotation, to amplify labeled data and mix it with unlabeled data and their pseudo-labels, where the pseudo-labels are regarded as hard labels. We control the ratio of labeled data to unlabeled data at 5:1. Finally, we fine-tune the pre-trained model again on this merged dataset to obtain a better model.

## 3 Experimental Settings

### 3.1 Datasets

### 3.1.1 Public Datasets

**LIVECell**: LIVECell [34] is a manually annotated and expert-validated dataset of phase-contrast images, consisting of 5,239 images with 1,686,352 cells from a diverse set of cell morphologies.

Table 1: Development environments and requirements.

| System | CentOS |
|---|---|
| CPU | Intel(R) Xeon(R) CPU E5-2630 v4 @ 2.20GHz |
| RAM | 8×32GBs |
| GPU | 3×NVIDIA GeForce RTX 3090 |
| CUDA version | 11.4 |
| Programming language | Python 3.8 |
| Deep learning framework | PyTorch (Torch 1.10, torchvision 0.11.0) |
| Specific dependencies | mmdetection==2.19.0, mmsegmentation==0.20.2 |

Table 2: Experiment environments for segmentation efficiency.

| System | Windows |
|---|---|
| CPU | Intel(R) Core(TM) i5-12600KF 3.70 GHz |
| RAM | 4×8GBs |
| GPU | 1×NVIDIA GeForce RTX 3060 |

**Sartorius competition data**: Sartorius [35] is a competition for detecting single neuronal cells in microscopy images. We used this competition's training set which includes 606 phase contrast microscopy images of neuronal cells.

### 3.1.2 Challenge Datasets

**Labeled training set**: 1000 manually labeled image patches from various microscopy types, tissue types, and staining types. Metadata (e.g., modality, tissue) for each image is not provided.

**Unlabeled training set**: 1712 image patches of different modalities, as well as 12 WSIs of immune fluorescent images.

**Tuning set**: 101 images with different sizes. It can be regarded as an official validation set, but the annotations are not available to competition participants. Participants can submit the inference results on the tuning set and get an F1 score of it.

**Validation set** A subset of labeled training set. We randomly divided the labeled training set into train/val by 0.8/0.2 to conduct hyperparameter tuning and validate the model performance.

**Testing set**: More than 200 images which are inaccessible to us.

## 3.2 Implementation Details

Our experiments were carried out on a computing cluster with three NVIDIA GeForce RTX 3090 GPUs for training. To assessing the applicability of our model, we performed all inferences on a personal computer equipped with an NVIDIA GeForce RTX 3060 GPU. Our implementation was based on the PyTorch framework [36], using the object detection library MMDetection [37] and the image segmentation library MMSegmentation [38]. For detailed information on the software and hardware, please refer to Table 1 and Table 2. Further information can be found in the source code we released on GitHub at https://github.com/Caixc97/nips_cellseg.

### 3.2.1 Training Protocols

In the detection part, we implemented various data augmentation strategies to improve the model's generalization performance during training. These included Mosaic, RandomAffine, MixUp, PhotoMetricDistortion, and RandomFlip. During training, patches were sampled with a window size of $1536 \times 1536$ and a stride of 768. For inference, we used the same window size and a stride of 1280. Images smaller than $1536 \times 1536$ were resized and padded to this size while maintaining their aspect ratio. We used an SGD optimizer with Nesterov momentum and an initial learning rate of 0.005/64, scaled by the batch size. The learning rate scheduler consisted of an exponential WarmUp and CosineAnnealing combination. The training settings for the preliminary and final models were the same, except for the total number of epochs (20 for the preliminary model and 15 for the final model). It has been reported that excessive data augmentation could produce many unrealistic images

Table 3: Training protocols of detection part.

| | |
|---|---|
| Network initialization | LIVECell [34]&Sartorius [35] pre-trained |
| Batch size | 4 |
| Patch size | 1536×1536×3 |
| Total epochs | 20 for preliminary model, 15 for final model |
| Optimizer | SGD with nesterov momentum ($\mu = 0.99$) |
| Initial learning rate (lr) | 0.005/64 |
| lr scheduler | WarmUp + CosineAnnealing |
| Training time | about 3 hours for preliminary model, 18 hours for final models |
| Loss function | IoULoss [39] + CrossEntropyLoss + L1Loss |
| Number of model parameters | 99.1M [32] |
| Number of flops | 281.9G [32] |

Table 4: Training protocols of segmentation part.

| | |
|---|---|
| Network initialization | LIVECell [34]&Sartoriuskaggle pre-trained |
| Batch size | 64 |
| Patch size | 64×64×3 |
| Total epochs | 10 |
| Optimizer | AdamW [40] |
| Initial learning rate (lr) | 6e-05 / 16 |
| lr scheduler | WarmUp + CosineAnnealing |
| Training time | about 1 hour |
| Loss function | CrossEntropyLoss + DiceLoss [41] |
| Number of model parameters | 29.06M [7] |
| Number of flops | 3.18G [7] |

and degrade the model's performance on real data [32]. To avoid this, we discarded Mosaic and MixUp during the final few epochs and added an additional L1 loss to prevent overfitting. The detailed training protocols for the detection part can be found in Table 3.

In the segmentation part, we used BoxJitter, RandomRotation, and RandomFlip for data augmentation. Patches were cropped from ground truth bounding boxes during training and predicted bounding boxes during inference. Bounding boxes with non-integer coordinates were rounded to the nearest. All patches were resized to $64 \times 64$ using nearest neighbor interpolation and restored after segmentation. We used an AdamW optimizer with an initial learning rate of 6e-5/64, scaled by the batch size. The learning rate scheduler consisted of an exponential WarmUp and CosineAnnealing combination. The detailed training protocols for the segmentation part can be found in Table 4.

To conduct hyperparameter tuning, we monitored the mean average precision (mAP) on the validation set during training. We then used the same hyperparameters to train on the entire dataset and obtain the model we submitted. Please note that in our final submission, we only conducted semi-supervised training with the detection part, as the segmentation part already provided satisfactory results.

## 4 Results and Discussion

### 4.1 Quantitative Results on Tuning Set

We evaluated our method's performance on the tuning set, which contains images of various cell types and imaging platforms, including some that are absent from the training set. Our approach achieved an F1 score of 0.8535, which is a significant improvement over the baseline method (F1 score of 0.5482). This result demonstrates that our method generalizes well to cell images from different modalities.

We also conducted ablation experiments to assess the effectiveness of incorporating unlabeled data in our approach. The results show that using unlabeled data lead to an F1 score improvement of 0.0513.

Table 5: Quantitative results on tuning set and ablation experiments.

|  | F1 score on tuning set |
|---|---|
| VSM | **0.8535** |
| - TTA | 0.8425 |
| - Training on unlabeled data | 0.7922 |
| - LIVECell [34]&Sartorius [35] pre-training | 0.7641 |
| Baseline | 0.5482 |

Table 6: Quantitative results on the validation set.

|  | F1 score on validation set |
|---|---|
| Baseline (end-to-end U-Net [7]) | 0.5770 |
| YOLOX-x + Swin-Transformer-t [42] | 0.8684 |
| YOLOX-x + U-Net [7] | 0.8729 |
| VSM (YOLOX-x + U-Net [7] + unlabeled data) | **0.8901** |

The number of unlabeled images used in the experiment was only slightly more than the number of labeled images. A larger amount of unlabeled data can lead to even more significant improvements in our approach.

We also conducted experiments on the situation of not using LIVECell and Sartorius data for pre-training and observed more significant performance degradation. It may indicate that labeled cell data, even from a different modality, can provide more effective performance improvement than unlabeled data. Transfer learning may be a more effective solution to the data problem we mentioned above. Hence, it is better to utilize both transfer learning and semi-supervised learning.

## 4.2 Qualitative Results and Visualization Results on the Validation Set

We evaluated our approach on the validation set, and the results are shown in Table 6. The superior performance of U-Net, which we observed in our experiments, was the deciding factor in choosing it as our segmentation backbone, instead of Swin-Transformer. Our approach's best F1 score was achieved by combining YOLOX-x and U-Net with the incorporation of unlabeled data (F1 score of 0.8901).

To investigate the generalization of our approach to cell images of different modalities, we visualized the model's prediction masks on the validation set, which contains various types of cells and imaging platforms. The visualization results, as shown in Figure 5, demonstrate that our approach has good generalization across modalities.

We also analyzed the failure cases on the validation set, as shown in Figure 6, and found that the model had difficulty recognizing cells that were not present in the training set in some cases (case 1 and case 2). Furthermore, the presence of red blood cells in the background of some images led to incorrect segmentation results (case 3). In some cases, the blurring of the image made it difficult to recognize cells (case 4 and case 5).

We recorded the accuracy of the validation set during training for both the preliminary and final models, as shown in Figure 7. We found that the final model's convergence was slower than the preliminary model, but it eventually converged to a higher precision. The plot also confirms the effectiveness of discarding excessive data augmentation during the final few epochs.

## 4.3 Segmentation Efficiency Results on the Validation Set

Based on the experimental results, the inference time can be calculated by this equation:

$$InferenceTime = 2.48 \times MillionPixels + 0.0037 \times NumInstance + 6.97 \quad (s) \quad (1)$$

Our segmentation efficiency results on the validation set are shown in Table 7, which indicates that our approach can efficiently segment cells of varying sizes.

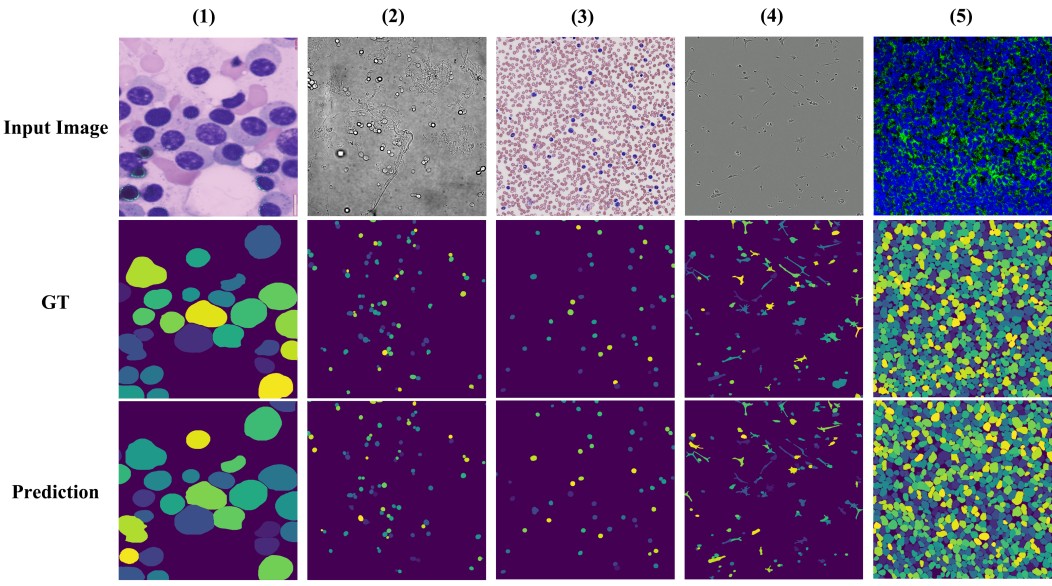

Figure 5: Some of the visualization results on the validation set.

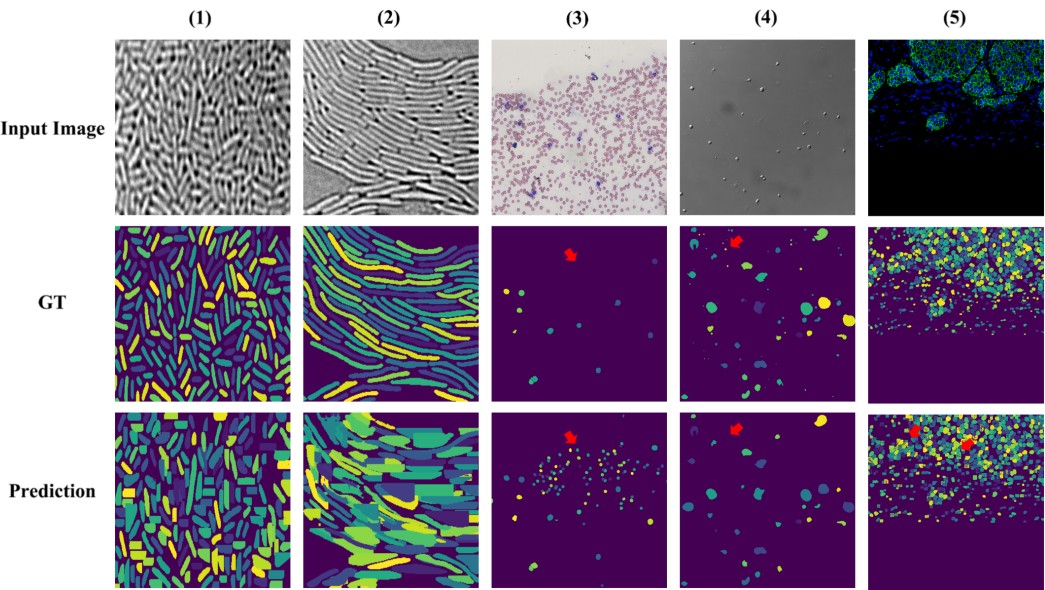

Figure 6: Failure case on the validation set.

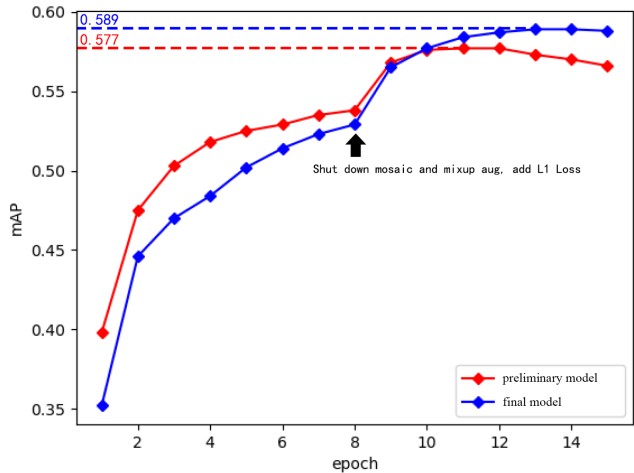

Figure 7: mAP of detection part on the validation set.

Table 7: Results of segmentation efficiency.

| Image name | Image size | Number of predicted instance | Inference time |
|---|---|---|---|
| cell_00003 | (480, 640) | 27 | 9.67s |
| cell_00008 | (480, 640) | 17 | 9.20s |
| cell_00012 | (1920, 2560) | 245 | 18.21s |
| cell_00017 | (3000, 3000) | 37 | 29.28s |
| cell_00037 | (944, 1266) | 133 | 10.46s |
| cell_00053 | (1024, 1024) | 873 | 11.65s |
| cell_00074 | (2048, 2048) | 6557 | 41.57s |
| cell_00077 | (1014, 1014) | 1656 | 9.67s |
| cell_00101 | (8415, 10496) | 76380 | 510.61s |

Table 8: F1 score on the testing set.

| | |
|---|---|
| Median F1-All | 0.8005 |
| Median F1-BF | 0.8894 |
| Median F1-DIC | 0.7692 |
| Median F1-Fluo | 0.1236 |
| Median F1-PC | 0.8747 |
| Mean F1-All | 0.6579 |
| Mean F1-BF | 0.8849 |
| Mean F1-DIC | 0.7088 |
| Mean F1-Fluo | 0.2435 |
| Mean F1-PC | 0.8196 |

Note that the time of startup time of docker is also included in the inference time. In practice, the actual inference time for each image can be reduced by 6.97 seconds (the constant in Eq 1).

## 4.4 Results on Final Testing Set

Table 8 depicts the outcomes of our proposed method on the final testing set, which demonstrates a high segmentation performance across the majority of cell image modalities. However, the performance of the method was observed to be relative low for some modalities, such as fluorescent images. Regrettably, further analysis of these outcomes is unattainable owing to the unavailability of the corresponding testing data.

### 4.5 Limitation and Future Work

Although our proposed method has achieved good results in experiments, there are still limitations that require further exploration:

Firstly, due to the scarcity of quantity and diversity of unlabeled data, we only utilized unlabeled data to generate pseudo labels for training. With sufficient data, many studies have demonstrated that representation learning with unlabeled data can effectively enhance model performance. Moreover, current methods mainly focus on designing universal tasks for learning image representation. In the future, we can investigate whether we can design unsupervised tasks that are tailored to the characteristics of cell images to learn better image representation.

Secondly, our ablation experiments reveal that the incorporation of transfer learning techniques improves model performance more significantly than the utilization of unlabeled data. Due to time constraints, we did not conduct more experiments with more datasets to test the effectiveness of transfer learning, which will be pursued in future work.

Thirdly, additional experimentation is required to choose the most appropriate backbone models for cell segmentation. In future work, we plan to involve conducting more experiments and exploring novel models specifically tailored for this purpose.

## 5 Conclusion

In this paper, we introduce VSM, a versatile semi-supervised model for multi-modal cell instance segmentation, which was evaluated in the NeurIPS 2022 Cell Segmentation challenge. Our quantitative experiments demonstrate the effectiveness of this method in addressing cell instance segmentation tasks across various tissues and imaging platforms, even when the amount of available data is limited. These findings suggest that in addition to collecting more high-quality labeled data, the proposed method has the potential to address the challenges of cell instance segmentation across different modalities.

## Acknowledgement

The authors of this paper declare that the segmentation method implemented for participation in the NeurIPS 2022 Cell Segmentation challenge utilized only datasets provided by the organizers and official external datasets and pre-trained models, without incorporating any private datasets. Additionally, the proposed solution is fully automatic, with no manual intervention.

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
