# OpenReview forum: "VSM: A Versatile Semi-supervised Model for Multi-modal Cell Instance Segmentation"
_NeurIPS.cc/2022/Challenge/CellSeg — Submitted to NeurIPS CellSeg 2022_

### Official Review · Reviewer_2sPy · 2022-12-17
**Paper contains the required information. But need to improve some parts for readability.**

**Rating:** 6
**Confidence:** 5

**Review:**

Overall, the contents include the required information suggested by the competition paper guideline. Instead of discussing the technical soundness or significance of the work, I focused only on the completeness of the paper. I found that the methodology description in the paper is complete enough, where one can grasp how to reproduce the result of VSM and understand how the method works. However, it still has far more room to be improved in the sense of readability. Here are some suggestions to improve the paper:

## Method Visualization
The visualization or illustration of the method and its results are over-simplified. I highly recommend to consider to redraw the method illustrations. You can more clearly illustrate the related modules, and how the method works in each step or phase.

## More Citations
Many sentences lack citations. I put some example sentences from the `Introduction` Section, but they should be carefully considered in the whole paper's contents.
- "Different from standard semantic segmentation and object detection, instance segmentation is detecting individual objects and segmenting them simultaneously."
- "The main challenge in this task is the diversity of image patterns and cell types."
-  "Blood cells in a bone marrow smear typically have rounded edges about 7 micrometers in diameter, while nerve cells have irregular edges and diameters."
- keywords that need related citations: semi-supervised learning, pseudo labeling, test-time augmentation, etc.


## Gentle Explanation
Please writhe the explanation more gently, especially for the `Experimental Setup` parts. Instead of just writing "See `Table x.x`" or "Link is here: `www.xxx`," the authors can provide the related information, not in the table or other considerations. This way, the readers can follow the instructions or be informed much better.


## Grammars/Expressions
The paper is generally well-written, but it needs thorough proofreading. Many obvious grammatical errors, wrong tense choices, and awkward expressions are spotted all over the paper.


## Minor Issues
- The paper contains some inconsistent hyphens. For example, "pre-trained," "fine-tuned," vs. "pretrained," and "finetuned" (in the figure).
- Missing footnotes for `Table 3`.
- The readers may not know the concept of the "tuning set." In this case, they could be confused when facing the performance reports on the "tuning set" and "validation set. Please clarify the definition of the sets in some parts of the paper.

---

### Official Review · Reviewer_iEUU · 2022-12-20
**Method is understandable, writing needs improvements**

**Rating:** 6
**Confidence:** 5

**Review:**

## Overall
The paper contains all required information to reproduce similar results and is generally well structured. I agree with Reviewer 2sPy on all his comments and remarks and will only comment on before unmentioned parts of the paper.

## Swin-Transformer for Segmentation
Table 6 reports results where the U-Net was substituted with a Swin-Transformer, but it is not mentioned in the text. The rationale behind trying and abandoning this design choice could be described in the text.

## Changes in Yolox training
The paper mentions that the Yolox training pipeline is changed by discarding Mosaic and MixUP augmentations and adding L1 loss in addition to the other two loss functions. The rationale behind this design choice is missing and could be mentioned in the text.

---

### Official Review · Reviewer_SRLJ · 2022-12-26
**paper is well structured and method is clearly described, more details are also needed**

**Rating:** 7
**Confidence:** 5

**Review:**

Basically the paper is well structured and has clearly described the implemented method and how to reproduce it. A novel two-stage instance segmentation is proposed to combine both yolox and unet. Just like other comments mentioned, i personally would recommend adding more explanations and ablations for the specific trade-off adopted.

# some details
- in 3.2.2 training Protocols, “Mosaic and MixUp are discarded, and additional L1 loss is added at the last few epochs”, it's better to explain why such strategies are adopted.
- in a two-stage instance segmentation is applied, object detection will be applied first, will some corner cases, e.g. pic 5 in figure 4 (many objects) have performance impact for the model? Does that InferenceTime also apply to eq. 1 in 4.3?

---

### Official Review · Reviewer_d3rN · 2022-12-27
**The paper is easy to follow, but still need improvement**

**Rating:** 6
**Confidence:** 4

**Review:**

Summary:

In this paper, the authors propose a semi-supervised cell instance segmentation model to address two main challenges in the Cell Segmentation area: (i) the poor multi-modality generalizability; (ii) the requirement for large amounts of labeled data. The experiment results have shown that the proposed VSM has high segmentation accuracy, generalization ability and efficiency.

Pros:
+ The paper structure is clear.

Cons:
+ The writing should be strengthened. There are some typos in the manuscript, e.g., Section 3.2 Line 1 'mmdection'. Besides, in Section 4.1, the authors write, 'We also conducted $ \cdots$', but in Section 4.2, they use  'We also conduct $\cdots$'

---

### Author Response · Authors · 2023-02-20
**Point to Point Response for the Reviewers**

We thank all reviewers for their constructive feedback and insightful comments. Our revised version of the paper showcases the major revisions as highlighted. Moreover, we conducted a thorough review of the paper and made numerous modifications throughout the paper to correct typos and improve overall readability. These modifications and rephrasing are not specifically highlighted.

## For Reviewer d3rN:
**Q1.1:** The writing should be strengthened. There are some typos in the manuscript, e.g., Section 3.2 Line 1 'mmdection'. Besides, in Section 4.1, the authors write, 'We also conducted ⋯', but in Section 4.2, they use 'We also conduct ⋯'

**A1.1:** We have conducted a thorough review of the paper and made numerous modifications throughout the paper to correct typos and improve overall readability.

## For Reviewer SRLJ:
**Q2.1:** in 3.2.2 training Protocols, “Mosaic and MixUp are discarded, and additional L1 loss is added at the last few epochs”, it's better to explain why such strategies are adopted.

**A2.1:** We added an explanation in 3.2.2 (underlined part).

**Q2.2:** in a two-stage instance segmentation is applied, object detection will be applied first, will some corner cases, e.g. pic 5 in figure 4 (many objects) have performance impact for the model? Does that InferenceTime also apply to eq. 1 in 4.3?

**A2.2:** Yes, eq. 1 applies in all cases, including WSIs and corner cases. The inference time is proportional to the object number. For cases with many objects, the performance impact is calculated by the item “0.0037 × NumInstance “

## For Reviewer iEUU
**Q3.1:** Swin-Transformer for Segmentation

**A3.1:** We discontinued the use of Swin-Transformer due to its subpar performance compared to U-Net in the experiment, as shown in Table 6. We added an additional explanation in 2.2.2 and 4.2 (underlined part).

**Q3.2:** Changes in Yolox training

**A3.2:** Please refer to A2.1.

## For Reviewer 2sPy:
**Q4.1:** Method Visualization

**A4.1:** We added additional illustrations in Fig. 3. It demonstrates each component of our method specifically.

**Q4.2:** more citations

**A4.2:** We checked the paper thoroughly and added citations, including but not limited to the examples you mentioned

**Q4.3:** gentle explanation

**A4.3:** We have greatly rewritten the experimental setup part and made some modifications in other parts to make our method easier to follow.

**Q4.4:** grammars/expressions

**A4.4:** See A1.1

**Q4.5:** minor issues

**A4.5:** The issues you mentioned have been addressed. We added the definition of each dataset in 3.1

---

### Decision · Program_Chairs · 2023-01-19

Accept